# Bridging the Gaps between Circulating Tumor Cells and DNA Methylation in Prostate Cancer

**DOI:** 10.3390/cancers13164209

**Published:** 2021-08-21

**Authors:** Bianca C. T. Flores, Margareta P. Correia, José G. Rodríguez, Rui Henrique, Carmen Jerónimo

**Affiliations:** 1Cancer Biology and Epigenetics Group, Research Center of IPO Porto (CI-IPOP)/RISE@CI-IPOP (Health Research Network), Portuguese Oncology Institute of Porto (IPO Porto)/Porto Comprehensive Cancer Center (Porto.CCC), Rua Dr. António Bernardino de Almeida, 4200-072 Porto, Portugal; bianca.troncarelli@ipoporto.min-saude.pt (B.C.T.F.); margareta.correia@ipoporto.min-saude.pt (M.P.C.); henrique@ipoporto.min-saude.pt (R.H.); 2Department of Pathology and Molecular Immunology, School of Medicine & Biomedical Sciences, University of Porto (ICBAS-UP), 4050-513 Porto, Portugal; 3Circulating Tumor Cells Group, A.C.Camargo Cancer Center, São Paulo 01508-010, Brazil; joserodriguez@misena.edu.co; 4Department of Pathology, Portuguese Oncology Institute of Porto (IPOP), 4200-072 Porto, Portugal

**Keywords:** prostate cancer, circulating tumor cells, DNA methylation

## Abstract

Prostate cancer is the second most common male malignancy, with a highly variable clinical presentation and outcome. Therefore, diagnosis, prognostication, and management remain a challenge, as available clinical, imaging, and pathological parameters provide limited risk assessment. Thus, many biomarkers are under study to fill this critical gap, some of them based on epigenetic aberrations that might be detected in liquid biopsies. Herein, we provide a critical review of published data on the usefulness of DNA methylation and circulating tumor cells in diagnosis and treatment decisions in cases of prostate cancer, underlining key aspects and discussing the importance of these advances to the improvement of the management of prostate cancer patients. Using minimally invasive blood tests, the detection of highly specific biomarkers might be crucial for making therapeutic decisions, determining response to specific treatments, and allowing early diagnosis.

## 1. Introduction

Prostate cancer (PCa) is the second most commonly diagnosed male malignancy, with an estimated 1,414,259 new cases detected worldwide in 2020 [1], as well as the third leading cause of cancer mortality among men, with an estimated 375,304 deaths occurring in the same period [1]. Although novel therapies with proven benefits have been developed in recent years [2,3], increases in survival rates are meager. PCa is a very heterogeneous disease, ranging from indolent, which eventually delays diagnosis, to aggressive disease, which metastasizes and causes significant morbidity and lethality [4].

When organ-confined, PCa is mostly curable through radical surgery or radiotherapy. However, locally invasive or systemic disease remains incurable, although control can be achieved through androgen deprivation therapy (ADT), eventually complemented with radiotherapy [5,6,7]. Nonetheless, 10–20% of patients with metastasized PCa develop castration-resistant disease (CRPC) within 5 years, with a median survival of only 14 months [8,9,10].

There are several mechanisms involved in the emergence of CRPC, many of which involve the androgen receptor (AR), including receptor amplification, activating mutations, constitutively active truncated splice variants, phosphorylation, and methylation [11]. In particular, AR-V7 overexpression has been associated with increased risk of disease recurrence after radical prostatectomy in hormone-naïve prostate cancer patients [12]. Furthermore, constituents of the AR complex, including epigenetic mediators, may be overexpressed (co-activators) or repressed (co-repressors), and other signaling pathways may also be activated, including the MAPK, PI3K/Akt, and Wnt pathways [13,14,15].

Next-generation hormonal therapies, such as the CYP17A1 inhibitor abiraterone, which impairs the androgen synthesis pathway, or the AR antagonist enzalutamide, are options for metastatic CRPC (mCRPC); nevertheless, acquired resistance usually arises within 2 years [16,17], and none of these treatments are curative [10], reinforcing the urgent need for new therapeutic approaches. Thus, while effective biomarkers for predicting PCa aggressiveness are required to avoid overtreatment [18], equally effective biomarkers are needed to help define the best therapeutic strategy for advanced disease [19,20].

The evaluation of target genes’ status using immunohistochemistry, fluorescence in situ hybridization (FISH), and other methods performed in tissue samples of the primary tumor remains the cornerstone of therapeutic decision making. However, tumor cells evolve over time, not only because of genomic instability, but also under pressure from the immune system and therapeutic interventions, increasing tumor heterogeneity. Moreover, metastases usually acquire molecular features that differ from the primary tumor, making them a less reliable source of information for guiding clinical strategies. Therefore, it is imperative to develop biomarkers that might be assessed using non- or minimally invasive techniques, enabling the real-time follow-up of minimal residual disease, recurrence, and metastization, as well as therapy-resistant clonal selection within tumor cell populations [21]. Liquid biopsies comply with most of these requirements.

## 2. Liquid Biopsies

Very early in the formation and development of a primary tumor, cells might be released into the bloodstream. These circulating tumor cells (CTCs) are usually scarce, especially during the earliest stages of cancer development, but they can be enriched via different technologies, taking advantage of their physical and biological properties [22]. The real-time analysis of CTCs using liquid biopsies is feasible and may aid in disease monitoring [23]. The importance and relevance of CTCs in cancer research can be observed by the increasing number of publications on this subject (more than 26,000 published articles were found in a PubMed search performed on 12 July 2021). In addition to CTCs, the analysis of circulating cell-free tumor DNA (ctDNA)—released from tumor cells undergoing apoptosis or necrosis [24]—also represents a fast, reliable, cost-effective, and minimally invasive approach [25] for the real-time monitoring of cancer evolution, better representing the heterogeneous profile of all tumor subclones [25,26]. The improvement of sensitive molecular assays enables the screening of ctDNA for tumor-specific aberrations; consequently, ctDNA and CTC assessment have become a competing source of biomarkers [27]. Nonetheless, from a broader perspective, the information acquired from both sources (CTCs and ctDNA) is complementary and might be selected according to the type of analysis required [28].

As previously stated, tumor tissue samples might not adequately represent tumor heterogeneity, precluding accurate outcome prediction and treatment efficacy [29,30]. Furthermore, depending on the tumor’s anatomical location and the patient’s physical condition, obtaining a tissue biopsy might be unfeasible [31,32]. In this context, liquid biopsies obtained from easily assessable body fluids, such as blood, urine, or sputum, have emerged as a promising alternative to cover these needs [33].

Importantly, researchers from this field have combined their efforts and share the scope of their work, disseminating tools and data, which has enabled further progress to be made [34]. However, the small amount of genetic material derived from the CTCs and ctDNA may limit the use of liquid biopsies in cancer patients, as it is not always possible to obtain large volumes of blood. Furthermore, there is a need for the standardization of preanalytical variables and isolation procedures. In this context, global consortia (Cancer-ID in Europe and BloodPAC in the United States, for example) are pivotal in the standardization of liquid biopsy-based methods [35].

### 2.1. Circulating Tumor Cells

The half-time of CTCs in the bloodstream is rather short (1–2.4 h) [36], and the process of release into the bloodstream remains controversial, whether or not it is predetermined. Nevertheless, conditions in the bloodstream are severe for epithelial tumor cells, and CTCs likely undergo a strong selection process [37]. Indeed, this is consistent with the frequent presence of apoptotic and fragmented CTCs in the peripheral blood of cancer patients [38].

The dynamics of the metastatic process have been the focus of intense research over the last two decades. Consequently, it has been found that tumor cells may disseminate, even when the tumor is still “confined”, or before the detection of the primary tumor by imaging [39,40]. Moreover, the assessment of living tumor cells has the advantage of directly measuring the response to treatment compared with evaluation after tissue fixation [41,42]. Notwithstanding the potential of CTCs, their use is limited by their scarcity and the need for highly specialized techniques enabling their isolation.

Circulating tumor cell isolation assays usually start with an enrichment step using different techniques. In principle, CTCs may be positively or negatively isolated based on biological (i.e., the expression of protein markers) or physical (i.e., size, density, deformability, or electric charges) properties. This may also be accomplished through a combination of physical and biological properties in a single device [43] (Figure 1).

Several methods based on physical properties have been developed, allowing for CTC separation without surface markers. Examples include centrifugal density gradient (Ficoll, OncoQuick™) [44] and filtration with a special filter/membrane (ISET^®^, a tumor cell size isolation method developed by Rarecells, France). CTC levels detected by ISET^®^ were correlated with imaging findings, and patient disease progressed within one month after an increase in CTC counts [42] (Table 1).

Nonetheless, the use of microfluidic devices is increasing, allowing for the improvement and standardization of CTC enrichment methods. Vortex technology allows for CTC capture with a high purity, efficiency, and speed through laminar microscale vortices, isolating and concentrating CTCs from blood. In this case, CTC capture is based on cell size, shape, and deformability. Vortex technology enabled the identification of CTCs in 80% of advanced-stage PCa patients, among which 11.5% did not express epithelial markers [45].
cancers-13-04209-t001_Table 1Table 1CTC enrichment methods, organized by technology type.
Enrichment Subcategory

Technology

Selection Criteria

Main Features

References
**Immunoaffinity—Positive selection**CellSearchEpCAM and Pan-CK positive selectionFDA-Approved[21,46,47]AdnaTestAntibody cocktailImmunomagnetic selection, followed by RT-PCR[48]MACSEpCAMMagnetic beads for positive selection through EpCAM[49]MagSweeperEpCAMHigh purity, 9 mL/h[50]CTC-ChipEpCAM1–2 mL/h[51]**Immunoaffinity—Negative selection**EasySep Human CD45 Depletion KitCD45Easy-to-use, high-throughput[52]MACSCD45Immunomagnetic selection[53,54]**Biophysical**RosetteSep CTC Enrichment CocktailDensity, Antibody CocktailImmunoaffinity assay, centrifugation[55]OncoQuickDensity, SizeIsolation by intense centrifugation[56]Ficoll-PaqueDensityCheap, easy-to-use, centrifugation[57]ISETSize, DeformabilityFixed samples in membrane[58,59,60]ScreenCellSize, DeformabilityCheap, easy-to-use, membrane[61,62]ParsortixSize, DeformabilityViable cells retained by size[63,64,65]VortexSizeNo RBC lysis required, captures viable cells in suspension, easy-to-use[45,66]DEPArrayElectrical SignatureRequires pre-enrichment, allows recovery and manipulation of viable cells[67,68,69]**Functional Assays**EPISPOTProtein secretionDiscriminates between viable and apoptotic CTCs using protein secretion[44,70]

Another example of label-free enrichment is the Parsortix System, intended to capture rare cells, which is based on patented microfluidic particle separation technology and relies on a very strict and repeatable technique. Its single-use separation cassettes allow for the subsequent culture and characterization of cells of interest, which are captured based on their size and resistance to compression [63].

Conversely, other methodologies have been developed to capture CTCs based on their surface markers. An immunomagnetic enrichment device called MagSweeper captures CTCs from samples using magnetic rods covered with removable plastic sleeves. These sleeves enable multiple capture and release cycles, thereby assuring their high purity and capture efficiency. For example, CTCs in patient blood samples can be isolated with an almost 100% purity and 60% capture efficiency [50].

CellSearch technology has been cleared by the Food and Drug Administration (FDA) for CTC isolation and enumeration from patients with metastatic breast, prostate, and colorectal cancer (mBC, mPC, mCRC) and has been on the market for more than 15 years. The enrichment of CTCs is still predominantly conducted using EpCAM (Epithelial Cell Adhesion Molecule), complemented by the standard detection of pan-keratin, CD45, and DAPI, allowing the further characterization of CTC subpopulations [21].

The morphology of CTCs may vary depending on the origin of the primary tumor, and its frequency is usually 1 or fewer CTCs per 106–107 leukocytes, depending on the disease stage and aggressiveness [71,72]. In addition, CTCs may be found in circulation as single cells or clusters, which bear a higher metastatic potential [73]. Once isolated, CTCs can be quantified and characterized at the molecular level to further our understanding of cancer biology, as well as being tested as biomarkers, with potential application in clinical settings [74,75]. Indeed, following CTC isolation, several techniques can be used to investigate gene and protein expression [58]; genomic profiling can be carried out by sequencing; functional experiments can be conducted to evaluate metastasis, cell–cell communication, drug testing, and many other experiments [76] (Figure 1).

Importantly, CTCs can be maintained in culture in vitro, either dissociated or as organoids, in which case they may be maintained for at least six weeks, as demonstrated by Mout et al. [41]. Establishing PCa cell lines from liquid biopsy samples provides several advantages, including a lack of contaminant (and competing), normal epithelial and stromal cells, as well as the possibility of obtaining metastatic samples from patients with bone disease in a minimally invasive manner [41].

### 2.2. Circulating Tumor Cells in Prostate Cancer Patients

Although CTC quantification using the CellSearch system, cleared by the FDA for metastatic prostate cancer [77], was found to be superior to the use of serum PSA for predicting overall survival in that setting [78], the clinical relevance of CTC enumeration in nonmetastatic PCa remains unclear [79].

The first study carried out to investigate gene expression changes in CTCs during CRPC development evaluated paired samples from 29 patients before ADT and at disease relapse. A panel of 47 genes related to PCa progression was assessed by qPCR, and it was demonstrated that CTCs are also informative regarding therapy response in a metastatic disease setting. Moreover, the *MDK* gene expression in CTCs was associated with poorer prognosis among metastatic PCa patients, emphasizing the importance of CTC gene profiling in complement to CTC enumeration, and adding relevant information concerning prognosis and treatment response [80].

In another report, high *ZEB1* expression in CTCs after one cycle of docetaxel was associated with poorer outcomes, further demonstrating its value as a biomarker with clinical application in cases of CRPC. Importantly, *MYCL* overexpression was detected even in a set of samples with less than 5 CTCs per 7.5 mL of blood [81].

Additionally, one study that evaluated differentially expressed genes in paired samples before and after surgery/radiotherapy showed no differences in CTC counts (74.1% vs 66.6%). However, although EMT markers were only expressed in 7% of patients’ CTCs before therapy, they were expressed in 63.0% of CTCs after therapy. Stem cell markers were also evaluated in patients’ CTCs before surgery/radiotherapy [79]. Overall, detection systems based only on epithelial-cell surface markers, such as EpCAM, and cytoskeletal proteins, such as CKs (Cytokeratin), are not ideal for the characterization of all CTC subpopulations [79,82], which is important in order to fully assess tumor heterogeneity.

A multicenter study enrolling 118 men demonstrated that patients with at poor-risk of mCRPC and whose CTCs’ androgen receptor splice variant 7 (AR-V7) status was positive did not benefit from abiraterone or enzalutamide therapy but could still benefit from docetaxel or cabazitaxel treatment [83]. Interestingly, these findings confirm previous data on the association of AR-V7-positive patients’ sensitivity to taxane-based chemotherapy [84,85,86].

Furthermore, Salami et al. detected CTCs in 33 out of 45 patients with localized PCa, demonstrating the ability to isolate and characterize CTCs morphologically and genomically even in early-stage disease. Furthermore, a high *AR* expression in those cells was associated with biochemical recurrence (defined as a PSA of 0.2 ng/mL or greater) and metastatic progression in patients submitted for radical prostatectomy [87]. These results are rather encouraging, considering the large number of tissue-based molecular markers under evaluation for PCa diagnosis and prognosis [88,89,90,91], and that might be less clinically informative considering that they mostly examine cells with invasive (but not necessarily metastatic) properties.

Because epigenetic alterations pervade the whole spectrum of cancer initiation and progression [92,93], the characterization of epigenetic aberrations in CTCs might provide an additional set of clinically relevant information [94,95].

## 3. DNA Methylation in Prostate Cancer Liquid Biopsies

Although the search for efficient biomarkers in oncology has been mostly focused on genetic mutations, their application as diagnostic biomarkers is challenged by the wide variety of those alterations, even for the same gene [96]. On the other hand, epigenetic modifications are more stable, are largely restricted to gene promoter regions, and maintain specific patterns within the same cancer model, supporting their investigation in the context of cancer biomarker development [58,97,98].

DNA methylation was the first epigenetic modification to be identified in cancer and is currently the most studied [99,100]. It involves the addition of a covalent methyl group, donated by S-adenosylmethionine (SAM), to the 5-position carbon of a cytosine ring to form 5-methylcytosine (5mC) [101,102]. This mechanism is catalyzed by DNA methyltransferases (DNMTs), specifically DNMT3a and DNMT3b, which actively promote de novo DNA methylation during embryonic development, generating a tissue-specific DNA methylation. Conversely, DNMT1 is often associated with the maintenance of pre-existent methylation patterns during subsequent replications (Figure 2) [102]. Usually, this process affects cytosine residues at CpG dinucleotides, some of which are clustered in so-called CpG islands, which are commonly located at the 5′ region of genes and are present in 60% of human gene promoter regions [99,102,103].

The excessive methylation (hypermethylation) of the promoter region often results in repression of the nearby gene. Nonetheless, depending on the localization of DNA methylation, this mechanism may result in different effects [101,104]. Epigenetic gene silencing by DNA promoter methylation may occur directly, through transcription factors impeding the binding to target sites, or indirectly, through methyl-CpG-binding proteins (MBP). The latter act by recruiting other enzymes, such as DNMTs and histone deacetylases (HDAC), leading to chromatin conformation alterations that further suppress gene transcription [99,101].

In 2003, Peter Laird, writing about recent advances in DNA methylation, postulated that it would become a powerful biomarker for cancer diagnosis. Indeed, DNA methylation holds several key properties required for biomarker development: easy detection through well-standardized techniques, stability in formalin-fixed samples over time, presence in various bodily fluids, and cell-type specificity [105]. Methodological and experimental obstacles are the major causes of the delay in the clinical implementation of DNA methylation-based biomarkers derived from basic and translational research. From more than 12,000 scientific papers describing new targets, only a few were tested in clinical trials, and three were approved [106,107] as a valuable assessment of occult disease risk in men with negative prostate cancer biopsy: *GSTP1*, *RASSF1*, and *APC*.

Techniques used for DNA methylation detection fall into three main groups: bisulfite conversion-based methods, restriction enzyme-based approaches, and affinity enrichment-based assays. Currently, bisulfite conversion-based methods are the most commonly used. Nonetheless, choosing the best method depends on several variables, such as the specific biological problem, the resolution required, the available instruments, and the associated costs [108].

In PCa tissue, Strand et al. described the candidate methylation markers *PITX2*, *C1orf114* (*CCDC181*), and the GABRE~miR-452~miR-224 locus as independent predictors of biochemical recurrence, in addition to the three-gene signature *AOX1/C1orf114/HAPLN3*, demonstrating the potential of DNA methylation biomarkers for PCa management. Nonetheless, all these biomarkers have been assessed in tissue specimens only [109].

Wu et al. identified an AR-MethSig covering 1000 genomic regions in metastatic CRPC circulating tumor DNA and was able to identify a subgroup of more aggressive tumors with hypomethylation at putative *AR* binding sites [110]. Previous studies disclosed poorer outcomes for patients with *AR* overexpression in plasma [111,112], uncovering the innovative connection between liquid biopsy and DNA methylation; both are promising tools at the service of effective and feasible blood-based tests for use in cancer diagnosis, prognosis, and therapy monitoring.

## 4. DNA Methylation in Prostate Cancer CTCs

Friedlander et al. evaluated CTCs enriched by a method that relies on the biological proclivity of tumor cells to invade collagenous matrices and that allows for their identification independently of EpCAM status and their propagation in culture. Genes that already exhibit an abnormal methylation and copy number in metastatic CRPC tumor tissue [113] were now evaluated in CTCs. A number of different candidates were found to be methylated, including genes critical to androgen synthesis, metabolism, and signaling, such as *CYP11A1*, *CYP11B1*, *CYP17A1,* and *CYP19A1* [114].

One of the main applications of DNA methylation analysis in CTCs derives from important information concerning their molecular and biological nature, originating in the cell–cell communications within the microenvironment. Interestingly, another study compared, for the first time, *GSTP1* and *RASSF1A* methylation in EpCAM-positive CTCs and exosomes from the same blood draw. *GSTP1* and *RASSF1A* were highly methylated both in EpCAM-positive CTCs and paired plasma-derived exosomes, and *GSTP1* methylation was significantly correlated with low overall survival in EpCAM-positive CTCs [115].

AR-V7 expression in CTCs was previously shown to predict resistance to new generation anti-AR-targeted treatments (abiraterone and enzalutamide), but not to taxane-based chemotherapy in metastatic CRPC [84,85,116]. The improvement of molecular assays to detect AR-V7 with a high analytical sensitivity, specificity, and accuracy is critical for its use in clinical practice [117,118,119]. Recently, Sharp et al. showed that patients with CTCs not detectable by the AdnaTest method often demonstrate the isolation of CTCs on the CellSearch platform and express AR-V7 protein that matches the tumor tissue [120], highlighting the importance of the detection method and the gene expression concordance between tumor tissue and CTCs.

## 5. Conclusions

Liquid biopsies based on serial and minimally invasive blood tests have the potential to detect tumor progression in real time by extracting molecular information from CTCs. Meanwhile, the detection of biomarkers in CTCs might be advantageous for therapeutic decisions, especially if CTCs are indicative of response to specific treatments and could aid in early diagnosis.

The FDA has already approved CTC assessment methods for clinical use. Although CTC quantification is described as a prognostic marker associated with survival, the molecular characterization of CTCs has the potential to offer more accurate information for the monitoring of treatment response, overcoming potential limitations due to tumor heterogeneity. Many biomarkers are currently under study in PCa patients’ liquid biopsies, and the future of these precision oncology initiatives will rely on the feasibility of identifying different molecular tumor subtypes, enabling improved diagnosis, monitoring, and treatment at all disease stages.

## Figures and Tables

**Figure 1 cancers-13-04209-f001:**
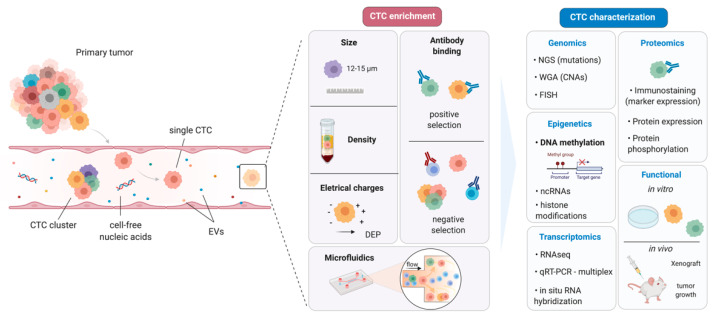
CTC enrichment methods and characterization. CTCs can be enriched on the basis of their physical or biological properties, such as size, density, electrical charge, antibodies, and/or the use of microfluidic devices. After enrichment, several methods can be applied to characterize the various subgroups of CTCs, using well-known technologies, including methylation analysis of target genes. CTC: Circulating Tumor Cell; EVs: Extracellular Vesicles.

**Figure 2 cancers-13-04209-f002:**
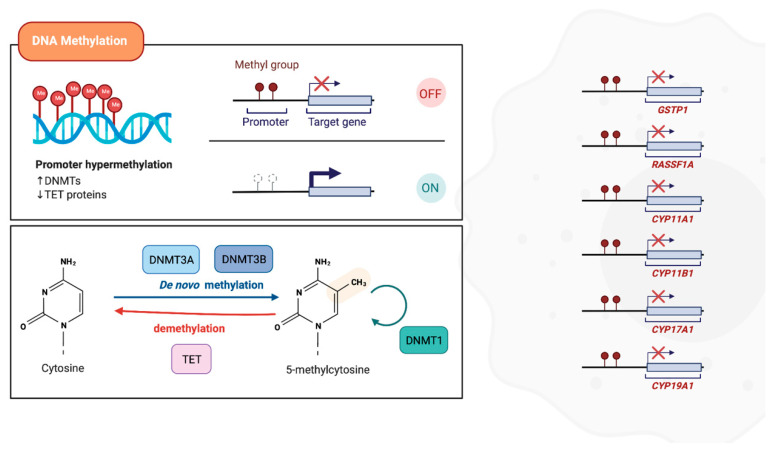
Relevant hypermethylated genes in prostate CTCs. Methylation is characterized by the addition of a covalent methyl group. This mechanism is catalyzed by DNA methyltransferase enzymes (DNMTs), while TET proteins promote a locus-specific reversal effect of DNA methylation. Herein, the targets already found to be hypermethylated in prostate CTCs are also represented.

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
