# Peer review of "Bridging the Gaps between Circulating Tumor Cells and DNA Methylation in Prostate Cancer"

_cancers, 2021, doi:10.3390/cancers13164209_

Round 1
Reviewer 1 Report
Reviewer's report
Title: Bridging the gaps between Circulating Tumor Cells and DNA methylation towards precision medicine in Prostate cancer
Version: 1 Date: 2021 August 7th
Reviewer's report:
In this manuscript, the Authors reviewed current biomarkers of prostate cancer in circulating tumor cells and DNA methylation. With respect to a previous paper published on Cells by the same Authors (see PMID 32150897), about the same topic but on the four major cancer types (Lung, breast, colorectal, and prostate), they now focused their review only on prostate cancer.
Overall, this manuscript is well written but for many topics they could deepen their descriptions.
Major points:
Introduction, line 65: The concept: “very early, during the formation and development of a primary tumor, cells are released into the bloodstream.” Is not completely correct. In early stages it is very difficult to find ctDNA biomarkers in liquid biopsy, in fact for instance Liu MC et al PMID 33506766, found close to 0% sensitivity in stage I and II for prostate cancer. Please change this sentence and cite this important paper on the field.
It should be important to add a paragraph in the section “3. DNA methylation” in which the Authors summarize briefly, eventually with a picture, the methods to assess the DNA methylation status, i.e. bisulfite NGS, methylation array, bisulfite real time PCR, Oxford Nanopore Technologies.
There are no reports on urine as a possible source of liquid biopsy specimen. Please add a section of what has been published on this, since it has been shown to be very important in prostate cancer.
Line 249: AR-V7 expression has only been briefly mentioned. I think it should be described in more detail.
A paragraph regarding pitfalls, problems and factors affecting test performances in detecting circulating biomarkers in liquid biopsy must be added at the end of the manuscript (see for instance PMID 29297893, 31151158).
Author Response
Question / Comment 1. In this manuscript, the Authors reviewed current biomarkers of prostate cancer in circulating tumor cells and DNA methylation. With respect to a previous paper published on Cells by the same Authors (see PMID 32150897), about the same topic but on the four major cancer types (Lung, breast, colorectal, and prostate), they now focused their review only on prostate cancer.
Overall, this manuscript is well written but for many topics they could deepen their descriptions.
RE: We thank the Reviewer for the evaluation and for the positive comments. We would like to clarify that the paper previously published by the same group (same senior authors) in Cells, focused on ctDNA, whereas the present review specifically refers to DNA methylation and circulating tumor cells. Although there is some overlap, the manuscripts are, however, quite different.
Question 2. Introduction, line 65: The concept: “very early, during the formation and development of a primary tumor, cells are released into the bloodstream.” Is not completely correct. In early stages it is very difficult to find ctDNA biomarkers in liquid biopsy, in fact for instance Liu MC et al PMID 33506766, found close to 0% sensitivity in stage I and II for prostate cancer. Please change this sentence and cite this important paper on the field.
RE: We understand the Reviewer’s point and partially agree, as we are referring to the early release of circulating tumor cells into the bloodstream, and not ctDNA. In a previous paper published by Flores et al. (PMID: 31247977), we investigated CTCs in patients with non-metastatic rectal cancer and isolation of CTCs was accomplished in all cases, which sustains our statement. The sentence was modified (from “are released” to “might be released”, now line 67). Please note that Liu MC et al. paper (PMID: 33506766) refers to cell free DNA and not circulating tumor cells.
Question 3 / Recommendation. It should be important to add a paragraph in the section “3. DNA methylation” in which the Authors summarize briefly, eventually with a picture, the methods to assess the DNA methylation status, i.e. bisulfite NGS, methylation array, bisulfite real time PCR, Oxford Nanopore Technologies.
RE: We thank the Reviewer for this important suggestion. A paragraph addressing this issue was added (lines 244-249).
Question 4. There are no reports on urine as a possible source of liquid biopsy specimen. Please add a section of what has been published on this, since it has been shown to be very important in prostate cancer.
RE: We thank the Reviewer for allowing us to clarify this issue. The review is not intended to address ctDNA. Hence, we did not consider the addition of urine as a specimen to assess ctDNA in the manuscript.
Question 5. Line 249: AR-V7 expression has only been briefly mentioned. I think it should be described in more detail.
RE: We thank the Reviewer for the suggestion. As requested, we further elaborated on AR-V7 in the Introduction (lines 41-43).
Question 6. A paragraph regarding pitfalls, problems and factors affecting test performances in detecting circulating biomarkers in liquid biopsy must be added at the end of the manuscript (see for instance PMID 29297893, 31151158).
RE: We thank the Reviewer for this valuable comment and suggestion, which was incorporated (lines 91-95).
Reviewer 2 Report
This manuscript reviews circulating tumor cells and DNA methylation in the diagnosis and treatment of Prostate Cancer, and this review well covered these fields.
Circulating tumor cells and DNA methylation are described somewhat independently. Since this is not appropriate, the section title, 3. DNA methylation (line 202) could be “DNA methylation in prostate cancer liquid biopsy” for example. The title, simple summary, and sections 3 and 4 should be revised accordingly. The last paragraph of section 3. DNA methylation (lines 249 -257, description of AR-V7 expression in CTCs) could be moved to section 4. DNA methylation in prostate cancer CTCs.
In Fig. 2, prostate CTCs should be indicated in addition to the legend.
EpCAM (line 139) and CK (line 180) require brief explanations or full names.
Author Response
Question 1 / Recommendation. Circulating tumor cells and DNA methylation are described somewhat independently. Since this is not appropriate, the section title, 3. DNA methylation (line 202) could be “DNA methylation in prostate cancer liquid biopsy” for example. The title, simple summary, and sections 3 and 4 should be revised accordingly. The last paragraph of section 3. DNA methylation (lines 249 -257, description of AR-V7 expression in CTCs) could be moved to section 4. DNA methylation in prostate cancer CTCs.
RE: We thank the Reviewer for the suggestion. Accordingly, the subtitle was modified to “DNA methylation in prostate cancer liquid biopsies”. Furthermore, the description of AR-V7 in CTCs was moved to section 4. DNA methylation in prostate cancer CTCs (lines 283-285).
Question 2 / Recommendation. The rotational In Fig. 2, prostate CTCs should be indicated in addition to the legend.
RE: We thank the Reviewer for allowing us to clarify this issue. Figure 2 to refer to CTCs in general and not specifically to PCa derived CTCs. Thus, “CTCs” is appropriate in this setting.
Question 3 / Recommendation. EpCAM (line 139) and CK (line 180) require brief explanations or full names.
RE: We thank the Reviewer for drawing our attention to this. The full names were added.
Reviewer 3 Report
In this report, the authors gave an overview of how assessment of circulating tumor cells (CTCs), as a non-invasive approach, is applied in the clinical management of prostate cancer (PCa). They further focused on the aberrant DNA methylation identified in PCa CTCs that could serve as biomarkers. Overall, the review is well written and organized, and can be a good reference in the field.
Major concerns:
1. The authors gave a fantastic summary on the CTCs in PCa. However, there is a significant lack of review on the overall methylation part. In particular, there were only 3 literatures referred (115-117) in section 4. This maybe due to the scarce of studies on DNA methylation in PCa CTCs. One suggestion is the authors can also extent their focus by combining review of cfDNA, whose methylation has been studied more extensively.
2. As this review’s title runs “..towards precision medicine in prostate cancer”, however the authors did not provide enough reviews/examples on how detection of the aberrant DNA methylation could guide precision medicine for PCa patients.
Minor comment:
In figure 2, there is a duplicated schematic CYP19A1 gene.
Author Response
Question 1 / Recommendation. The authors gave a fantastic summary on the CTCs in PCa. However, there is a significant lack of review on the overall methylation part. In particular, there were only 3 literatures referred (115-117) in section 4. This maybe due to the scarce of studies on DNA methylation in PCa CTCs. One suggestion is the authors can also extent their focus by combining review of cfDNA, whose methylation has been studied more extensively.
RE: We thank the Reviewer for the positive comment about the manuscript. There is really a lack of studies evaluating DNA methylation in PCa CTCs and that explains the small number of cited references, as the Reviewer anticipated. We should mention that this was one of the reasons for choosing this subject as the main focus of this review, which we are also exploring in our research group. Regarding ctDNA, we have previously published a review (PMID: 32150897) focusing on methylation in ctDNA as a source of cancer biomarkers.
Question 2 / Recommendation. As this review’s title runs “.towards precision medicine in prostate cancer”, however the authors did not provide enough reviews/examples on how detection of the aberrant DNA methylation could guide precision medicine for PCa patients.
RE: We agree with the Reviewer. Thus, the title was modified to better fit the topics addressed in the review.
Question 3 / Recommendation. In figure 2, there is a duplicated schematic CYP19A1 gene.
RE: We thank the Reviewer for the detailed analysis of the manuscript. The figure was corrected.
Round 2
Reviewer 1 Report
The revisions made addressed almost all issues suggested in the first round.
Reviewer 3 Report
The authors have addressed all my concerns, and the manuscript has been improved.